# Tracheal Hemangioma Causing Lung Emphysema and Pneumopericardium in a Rabbit—A Case Report

**DOI:** 10.3390/ani12151907

**Published:** 2022-07-26

**Authors:** Małgorzata Kandefer-Gola, Kacper Żebrowski, Rafał Ciaputa, Wojciech Borawski, Eleonora Brambilla, Valeria Grieco

**Affiliations:** 1Department of Pathology, Division of Pathomorphology and Veterinary Forensics, Faculty of Veterinary Medicine, Wroclaw University of Environmental and Life Sciences, 50-375 Wroclaw, Poland; malgorzata.kandefer-gola@upwr.edu.pl (M.K.-G.); rafal.ciaputa@upwr.edu.pl (R.C.); 2Department and Clinic of Surgery, Faculty of Veterinary Medicine, Wroclaw University of Environmental and Life Sciences, 50-375 Wroclaw, Poland; wojciech.borawski@upwr.edu.pl; 3Department of Veterinary Medicine and Animal Science, University of Milan, Via dell’ Università 6, 26900 Lodi, Italy; eleonora.brambilla@unimi.it (E.B.); valeria.grieco@unimi.it (V.G.)

**Keywords:** rabbit, pneumopericardium, hemangioma, trachea, failure

## Abstract

**Simple Summary:**

Rabbits have become popular pets in recent years and thus are increasingly presenting as patients to veterinary clinics. The medical examination and diagnosis of this type of patient are difficult due to the high degree of sensitivity and weakness of evident symptoms. However, in pet rabbits, several tumors and pathologic conditions have been already reported. In the present case, a female 8-year-old pet rabbit showed a severe respiratory disorder and a lack of improvement after antibiotic therapy. A computer tomography scan was performed which revealed the presence of air in the pericardial sac, a pneumopericardium. The rabbit died from circulatory and respiratory failure soon after the examination, and the necropsy revealed the presence of a tumor histologically consistent with hemangioma within the lumen of the trachea. This tumor, reducing the lumen of trachea, caused the pet rabbit to have dyspnea, and most likely predisposed the rabbit to a pneumopericardium. Tracheal hemangioma and a pneumopericardium have never been described in a pet rabbit. Moreover, this is the first report on the association of the two pathologic conditions in an animal or human.

**Abstract:**

A pet rabbit (female, 8 years old, and mixed breed) with symptoms of dyspnea, apathy, and weight loss was treated for an acute respiratory infection. Due to the lack of improvement, it was referred to the Imaging Diagnostics Laboratory of the Department and Clinic of Surgery for a computer tomography scan of the thoracic cavity. The examination revealed the presence of air in the pericardial sac, a pneumopericardium, along with pulmonary emphysema. A few minutes after the examination, the rabbit developed circulatory and respiratory failure and died. Necropsy confirmed the presence of a pneumopericardium and pulmonary emphysema, and revealed, in the tracheal lumen, the presence of a tumor histologically consistent with hemangioma. A spontaneous pneumopericardium occurs when air from the respiratory system moves into the pericardial sac. This is the first case of the simultaneous occurrence of tracheal hemangioma and a pneumopericardium in a rabbit.

## 1. Introduction

A pneumopericardium is a rare condition defined as a collection of air, or other gases, within the pericardial cavity [1]. A pneumopericardium can lead to myocardial damage, cardiac tamponade, and death. In humans, the mortality rate in patients with a pneumopericardium is 57% [1]. Pneumopericardium is rarely described in animals, being reported in four dogs and one cat [2,3].

Respiratory hemangiomas are also rare in animals. So far, two cases of respiratory hemangioma have been reported in animals; one was located in the nasal cavity of a sheep and the other in the nasopharynx of a dog [4,5]. Independent of their location, hemangiomas occur most commonly in dogs, and in canines and other species, they appear as red to brownish well-delimited nodules. Histologically, the tumor, which is benign, is typically composed of variably sized spaces or lacunae filled with erythrocytes, lined by flattened cells closely resembling normal endothelium. Blood clots may also be occasionally visible in the lumen of the lacunar spaces [6].

The present report describes the first case of the simultaneous occurrence of tracheal hemangioma, which probably caused lung emphysema and pneumopericardium, in a rabbit.

## 2. History and Case Presentation

A domestic pet rabbit in middle age (female, 8 years old, and mixed breed) was referred to the Imaging Diagnostics Laboratory of the Department and Clinic of Surgery exhibiting dyspnea, apathy, anorexia, and slight weight loss. According to the information obtained from the owners, the rabbit was kept at home and fed a ready-made feed, well-balanced with nutrients. The owners claimed that the rabbit had never had disorders connected with the respiratory system. No data on a possible injury were available, and according to the information obtained, the animal was suspected of an acute respiratory infection. Therefore, the rabbit was subjected to seven-day antibiotic therapy with azithromycin at a dose of 20 mg/kg b.w (Sumamed, Teva Polska, Kraków, Poland). Additionally, 0.5 mg/kg b.w meloxicam (Metacam, Boehringer Ingelheim, Ingelheim, Germany) and nebulization with budesonide (Budixon Neb, Adamed, Pieńków, Poland) were given. Due to the lack of improvement, the patient was examined again, and since during auscultation increased respiratory noises were found, the animal was submitted to a respiratory system computed tomography (CT) scan to identify the cause of the current state. Before the CT, a complete blood count and biochemical exams were performed revealing mild monocytosis and leukopenia in the absence of other abnormalities.

For the CT, the animal was sedated with 0.1 mg/kg b.w medetomidine (Sedator; Dechra, Nortwich, United Kingdom) and 5 mg/kg b.w ketamine (Bioketan; Vetoquinol Biovet, Gorzów Wielkopolski, Poland) and intubated. During the contrast test (slice thickness time point: 0.6 mm/90 s), iomeprol (Iomeron, Bracco, Warszawa, Poland) was used at a concentration of 350 mg iodine/mL administered at a dose of 700 mg/kg b.w, intravenously. During the CT (nativ scan) of the nasal cavities, no abnormalities were observed, while the chest CT scan revealed features of emphysema, multifocal atelectasis, and mild diffuse thickening of the bronchial walls. No pulmonary lesions were detected. Of note, there was a small amount of gas in the pericardial sac (Figure 1), consistent with a pneumopericardium. Approximately 5 min after the end of anaesthesia, the animal suddenly developed respiratory and cardiac failure, and despite attempting cardiopulmonary resuscitation, it died. After the rabbit’s death, the owners considered performing a necropsy, and the body was frozen for 2 weeks in −4 °C. We have an awareness that freezing can alter cell structures; however, on the basis of our own experienc, we know that freezing did not have an impact on the ending diagnosis in the described case. After the owners’ consent, the rabbit cadaver was defrosted, and a complete necropsy was performed.

The external examination revealed only bloody discharge from both nostrils. In the thoracic cavity, the lungs were slightly congested, mosaic-like, and edematous, with scattered multifocal gross lesions consistent with emphysema and atelectasis. The internal examination of the trachea revealed, at the cranial level, the presence within the mucosa of a flat, elongated red/brownish lesion slightly extending to the caudal part of the larynx. In the pericardial sac, which was thin and stretched, no air was found at necropsy, probably due to the freezing and defrosting of the corpse. No cardiac-relevant lesions were detectable except for a dilation of the right ventricle. Within the abdominal cavity, a general congestion of the organs, consistent with circulatory acute failure, was the only pathological finding detected.

For histopathologic examination, sections of the trachea, lungs, and heart were collected, fixed in 7% buffered formalin, passed through graded alcohols, clarified in xylene, and paraffin-embedded. From the paraffin blocks, 4 μm-thick sections were obtained and stained with hematoxylin and eosin (HE) and Masson′s trichrome staining.

Histologically, the tracheal lesion revealed to be a well-demarcated tumor, composed of cystic spaces and lacunae, frequently filled by erythrocytes, separated by variably thick fibrovascular septa, and lined with a single layer of flattened cells, characterized by a moderate amount of faintly eosinophilic cytoplasm and elongated, thin, and hyperchromatic nuclei (Figure 2a). Cellular atypia, anisocytosis, and anisokaryosis were mild, and mitoses were <1 per high-power field (400×). Based on the histological features, the diagnosis of hemangioma was made. In the lung, edema, scattered areas of emphysema and atelectasis, and moderate lymphocytic peribronchial infiltrates were detected (Figure 2b). With Masson′s trichrome staining, no foci of lung fibrosis were recognizable.

## 3. Discussion

Primary neoplastic tumors within the trachea are rare in humans and animals as well [7,8]. The presence of a tumor within the lumen of the trachea may induce nonspecific clinical symptoms, such as cough and/or dyspnea. The exacerbation of symptoms occurs when a stress factor is activated, such as after exercise or eating [9]. In the presence of hemangioma, hemoptysis may be present (not in the present case), and in advanced cases, severe dyspnea may occur, and the cause of death is usually respiratory failure [4].

A pneumopericardium is a rare lesion, and its etiopathogenesis may be related to four different pathologic conditions: the existence of a fistula between the respiratory system and the pericardial sac, the action of a mechanical or pressure-altering trauma, the presence of an inflammation caused by gas-forming bacteria, or due to iatrogenic causes [10].

In case of development of a fistula connecting the respiratory system with the pericardial sac, the air comes out from the alveoli and accumulates in the pericardial sac with no possibility to return to the respiratory system. Such a fistula may arise because of the development of, inter alia, bronchial cancer or trauma within the chest [11]. In addition, in case of pleuropericardial tear, the air from the lungs may flow directly to the pericardial sac. When the visceral pleura is injured, air from the lungs may move through the perivascular and peribronchial spaces to the mediastinum, neck, retroperitoneal space, and pericardium, known as the “Macklin effect” [1,12].

Another path for pneumopericardium formation is represented by the transfer to the pericardial sac, through the blood or lymph, of gas-producing bacteria, such as *Clostridium perfringens* type A or *Klebsiella* [10].

A pneumopericardium can also develop on an iatrogenic basis; most often it is a complication of the endoscopic examination of the esophagus [13], and a case of iatrogenic pneumopericardium, as a complication of endotracheal laser surgery, has been described in a human patient [14].

Spontaneous pneumopericardium occurs, even more rarely, with the withdrawal of air from the alveoli into the pericardial sac [15,16,17]. In medicine, cases in which emphysema contributed to the formation of pneumopericardium are also described. In these latter cases, alveoli walls undergo rupture, and the air released may move through the perivascular spaces to the pericardial sac. The clinical symptoms are similar to those observed in tracheal neoplasm [16].

In the current case, the presence of the hemangioma reduced the tracheal lumen resulting in dyspnea, and most likely, in emphysema, which in any case cannot be excluded as pre-existing. The severe respiratory impairment led to the development of respiratory failure, and as a result, to the dilation of the right ventricle. Emphysema is defined as an excessive accumulation of air in the alveoli and interstitial tissue due to damage of the alveolar walls [18]. Probably, in the present case, due to the continuity of tissues, the air from the emphysematous pulmonary areas was transferred into the pericardial sac causing a pneumopericardium. In this process, the bronchial and peribronchial inflammatory reaction observed could have contributed to a more severe and extensive rupture of the alveolar walls with movement of air to the pericardial sac. The presence of the pneumopericardium contributed to the worsening of the clinical picture, and all processes led to impaired lung gas exchanges and circulatory impairment, which deteriorated after administration of the anesthetics, causing the animal’s death.

## 4. Conclusions

Spontaneous pneumopericardium has been reported several times in human patients [15,16]. In the present report, the first case of tracheal hemangioma, with concomitant pneumopericardium, was described in a rabbit. The pneumopericardium most likely established due to the co-existence of several predisposing factors, including tracheal hemangioma, emphysema, and moderate pulmonary inflammation.

## Figures and Tables

**Figure 1 animals-12-01907-f001:**
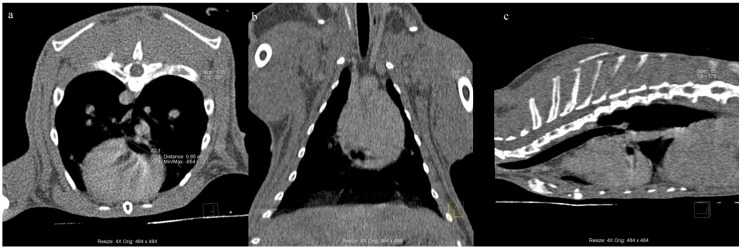
CT scan (native scan). A small amount of gas in the pericardial sac. Axial (**a**), coronal (**b**), and sagittal (**c**) reformat images of the chest computed tomography showing a small amount of gas in the pericardial cavity.

**Figure 2 animals-12-01907-f002:**
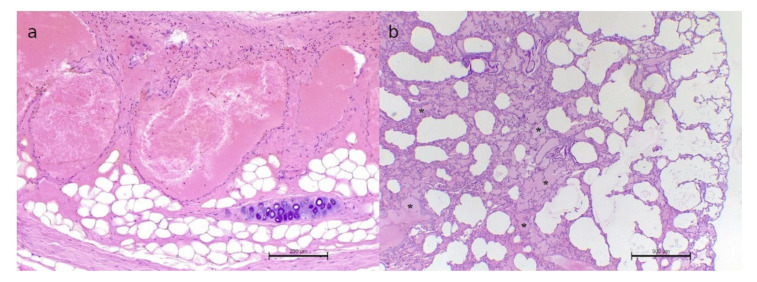
The histopathologic examination. (**a**). Tracheal hemangioma. Multiple vascular lacunae filled with erythrocytes and separated by stromal septa are visible. H&E staining. 100×. (**b**). Lung’s emphysema (empty spaces) and edema (asterisks) are recognizable and scattered in the pulmonary parenchyma. H&E staining. 40×.

## Data Availability

Not applicable.

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
