# Peer review of "Tracheal Hemangioma Causing Lung Emphysema and Pneumopericardium in a Rabbit—A Case Report"

_animals, 2022, doi:10.3390/ani12151907_

Round 1

Reviewer 1 Report

Dear author

This is a nice written Case report about rare described disease in animals

I had some problems to give my comments because there are no line number

Please see my comments

Location / your text

Title

Why you are not saying pneumopericardium in consequence of tracheal obstruction by an hemangioma, please see at the end an proposed title

End of introduction

Same as above

Chapter 2

exhibiting dyspnea

Is it possible to describe the type of dyspnoea

Chapter

I am missing typically vital signs heart rat respiration rate temperature

Chapetr 2

Question: an x-ray was not done by privat vet or the clinic

Chapter 2

meloxicam(Metacam

Please change to

“meloxicam (Metacam”

Chapter 2

2Question

Was there any stridor

Did the rabbit had dyspnoea at this time point

Chapter 2

Cardiac auscultation is missing

Chapter 2

“mild monocytosis and leukopenia”

Please give number and your laboratory reference

b.w

You are not writing b.w at any medication ?

Ct technique

Typically more details should be named like slice thickness time point of different runs depend to the contrast injection

CT description

I am missing results of the extrathoracic trachea

Moderate emphysema

Please describe your definition of moderate emphysema at the best with a reference

Did you measure the density of the lung

CT description

It is unclear which information is coming from native scan and which from contrast scan

after the test

Is this 5 minutes after the contrast application (during anaesthesia) or 5 minutes after end of anaesthesia

The internal ex-amination of the trachea revealed

Do you have a photo on this changes

Was the lumen narrowed by the tumor and if so, how much

In the lung, moderate edema

Please give a reference for your definition of moderate

End of chapter two

Was a bacterial culture of the pericardium done ?

Chapter 3

Emphysema / rabbit

Please give a reference that tracheal obstruction is causing lung emphysema in humans? animals ? rabbits ?

In this process, the bronchial and peribronchial inflammatory reaction observed could have contributed to a more severe and extensive rupture of the alveolar walls with movement of air to the pericardial sac

You should go with the same sentence to title and end of introduction

For example

“Tracheal hemangioma causing lung emphysema and pneumopericardium in a rabbit”

Reviewer 2 Report

Very interesting case report, well written and scientifically structured. Three minotr remarks:

Phrase: 'Due to the lack of improvement, the patient was referred again and, since during auscultation increased respiratory noises were found, the animal was submitted to a respiratory system computed tomography (CT) scan.'

Comment: Strange structure of the sentence, please rephrase. 

Phrase: 'Before the CT, complete blood count and biochemical exams were performed revealing a mild monocytosis and leukopenia in the absence of other abnormalities.'

Comment: Please add values and normal values.

Phrase: 'After the rabbit’s death, the owners considered performing a necropsy, and the body was frozen. After the owners’ consent, the rabbit cadaver was defrosted, and a complete necropsy was performed.'

Comment: Please add how long the body was stored and if and how this could influence the post mortem exam and possibly histopathology.
